# COVID-19 Vaccination Modifies COVID-19-Related Headache Phenotype: Evidence from Case–Control Study on 309 Participants

**DOI:** 10.3390/biomedicines13122900

**Published:** 2025-11-27

**Authors:** Henar Ruiz-Saez, Ana Echavarría Íñiguez, Yésica González Osorio, Javier Trigo López, Álvaro Sierra-Mencía, Andrea Recio-García, Álvaro Planchuelo-Gómez, Ana González-Celestino, María Luisa Hurtado, Leticia Sierra, Marta Ruiz, María Rojas-Hernández, Carolina Pérez Almendro, Marina Paniagua, Gabriela Núñez, Marta Mora, Carol Montilla, Cristina Martínez Badillo, Ana Guiomar Lozano, Cristina García-Iglesias, Ana Gil, Miguel Cubero, Ana Cornejo, Ismael Calcerrada, María Blanco, Ana Alberdi-Iglesias, César Fernández-de-las-Peñas, Ángel L. Guerrero Peral, David García-Azorín

**Affiliations:** 1Headache Unit, Department of Neurology, Hospital Clínico Universitario de Valladolid, Avenida Ramón y Cajal 3, 47003 Valladolid, Spain; hruizsaez@gmail.com (H.R.-S.); anaechavarria93@gmail.com (A.E.Í.); javiertrigolopez@gmail.com (J.T.L.); 2Health Research Institute of Valladolid (IBioVALL), Hospital Clínico Universitario de Valladolid C Rondilla Santa Teresa, 47010 Valladolid, Spain; ygoinvestigacion@outlook.com (Y.G.O.); alvarosierramencia@gmail.com (Á.S.-M.); andreareciogar99@gmail.com (A.R.-G.); cris6gar@gmail.com (C.G.-I.); 3Cardiff University Brain Research Imaging Centre (CUBRIC), Cardiff University, Cardiff CF103XA, UK; alvaro.planchuelo@uva.es; 4Imaging Processing Laboratory, Universidad de Valladolid, 47011 Valladolid, Spain; 5Valladolid East Primary Care Basic Health Area, 47010 Valladolid, Spain; anagonzalezcelestino@gmail.com (A.G.-C.); anaalberdi@hotmail.com (A.A.-I.); 6Valladolid West Primary Care Basic Health Area, 47012 Valladolid, Spain; marialhurtador@saludcastillayleon.es (M.L.H.); lsierram@saludcastillayleon.es (L.S.); martaruizg92@gmail.com (M.R.); mrojashdez@icloud.com (M.R.-H.); carolinaperezalmendro@hotmail.es (C.P.A.); marinapaniagua.mart22@gmail.com (M.P.); mariagabrielanunez9@gmail.com (G.N.); martamorasan@gmail.com (M.M.); carol.montilla29@gmail.com (C.M.); crismb92@gmail.com (C.M.B.); anitaguiomar092@gmail.com (A.G.L.); anagilc1990@gmail.com (A.G.); migc93@gmail.com (M.C.); acornejom@saludcastillayleon.es (A.C.); ismael.calcerrada@gmail.com (I.C.); m.blancogn@gmail.com (M.B.); 7Department of Physical Therapy, Occupational Therapy, Rehabilitation and Physical Medicine, Universidad Rey Juan Carlos, 28922 Alcorcón, Spain; cesar.fernandez@urjc.es; 8Headache Unit, Department of Neurology, Hospital Universitario Río Hortega, 47012 Valladolid, Spain; dgazorin@ucm.es

**Keywords:** COVID-19, COVID-19 vaccination, headache, secondary headache

## Abstract

**Background:** Headache is a common symptom during acute viral infections, and its pathophysiology has been linked with the immune response to the virus. Headache is one of the most frequent symptoms of coronavirus disease 2019 (COVID-19), and it has been associated with a more efficient immune response and a better prognosis. The aim of this article is to evaluate whether vaccination could modify the clinical phenotype and the probability of developing persistent headache after acute COVID-19. **Methods:** A case–control study comparing the duration of the headache and the clinical phenotype between fully vaccinated individuals and non-vaccinated controls was conducted. Each case was matched with two controls that were paired by age, sex, and prior history of headache. Patients were evaluated by a physician that administered a structured questionnaire and were followed up for at least three months. **Results:** The sample included 103 cases and 206 controls, with a median age of 42 (inter-quartile range (IQR) 33–51); 68% were female; and 26% had a prior history of headache. Headache had a shorter duration for vaccinated patients (4 (IQR 2–8) vs. 8 (IQR 4–16.5) days, *p* < 0.001). Vaccinated patients had a higher frequency of holocranial topography, pressing quality, phonophobia, and cranial autonomic symptoms. **Conclusions:** Our results suggest that full vaccination modifies the clinical phenotype of COVID-19 onset-associated headache and might lead to a shorter duration. These findings could represent an additional benefit of COVID-19 vaccines, which could extend to the post-COVID-19 phase and decrease the probability of a persistent disabling symptom such as headache.

## 1. Introduction

Headache is a frequent symptom experienced during the acute phase of Severe Acute Respiratory Syndrome coronavirus-2 (SARS-CoV-2) infection [1]. The International Classification of Headache Disorders (ICHD-3) requires the headache to develop in temporal ration to the onset of the infection, the significant worsening of the headache in parallel with the worsening of the infection, or an improvement or resolution in parallel with the improvement or resolution of the infection [2]. In cases of coronavirus disease 2019 (COVID-19), headache is one of the earliest symptoms, being present within 96 h from onset in 77% of patients [1]. The median duration of headache has been reported to be 7–14 days; however, in 9–19% of patients, it may persist three months after the acute phase of the disease, and in 16%, it may become chronic and remain present nine months after the acute phase of the disease [3].

The presence of headache as an onset symptom of acute COVID-19 has been associated with a better prognosis [4,5]. This observation has been linked with a more efficient immune response [6]. Evidence suggests that COVID-19 vaccines could be a protective factor for some chronic manifestations such as cough, headache, or arthritis [7]. No study has evaluated the differences in the duration and clinical phenotype of headache related to COVID-19 in vaccinated versus non-vaccinated patients. Accordingly, the aim of our study was to compare the clinical phenotype of COVID-19-associated headache between vaccinated and unvaccinated patients, in regard to (1) the duration of the headache over time, (2) the intensity of the headache, and (3) the frequency of associated symptoms. It was hypothesized that vaccination could change the clinical presentation of COVID-19-associated headache.

## 2. Materials and Methods

### 2.1. Study Design

An observational analytic study with a case–control design was conducted. This study was performed and reported in accordance with the Strengthening the Reporting in Observational Studies guidelines [8]. This study was approved by the Medical Research Ethics committee (PI 21-2500-TFG) and was conducted in accordance with the principles of the Declaration of Helsinki.

### 2.2. Study Population

The study population consisted of COVID-19 patients who presented with headache during the acute phase of the disease and were managed in an outpatient setting.

### 2.3. Study Setting

This study was performed in the Valladolid East Healthcare area, which has a population of 261,000 inhabitants. Twenty-two primary care facilities screened COVID-19 patients for eligibility.

### 2.4. Study Period

Cases were recruited between 1 January 2022 and 30 April 2022 and were followed up for a minimum period of three months. Controls were recruited between 8 March 2020 and 11 April 2020, during the first wave of the pandemic in Valladolid. Information regarding the control group has been partially published elsewhere [1], and the group was older, there were more female participants, and there was a higher frequency of a prior history of headache. To avoid potential biases, and to explore whether the differences between groups could be based on the vaccination status, each case was paired with two controls that were matched for age, sex, and prior history of headache. Matching was performed semi-automatically with the case–control matching function on SPSS, subsequently confirming the effectiveness of matching by performing statistical tests (chi-square for categorical or *t*-test for continuous variables).

### 2.5. Eligibility Criteria

The inclusion criteria were as follows: (1) a confirmed diagnosis of COVID-19 infection, either by polymerase chain reaction (PCR) or by IgM serum antibody testing; (2) age 18 years or older; (3) new-onset headache presented during COVID-19 that fulfilled the ICHD-3 criteria for 9.2.2.1 acute headache attributed to systemic viral infection [2]; and (4) agreement to participate and informed consent signature. Patients were defined as cases if they were fully vaccinated, according to the World Health Organization definition [9], or controls otherwise. Control patients had not received any dose of COVID-19 vaccines and had not been infected by COVID-19 before. The possible vaccines included the Pfizer BioNTech BNT162b2 vaccine, Moderna mRNA-1273 vaccine, Johnssons & Johnssons Janssen Ad26.COV2.S vaccine, and AstraZeneca ChAdOx-1s vaccine. Patients were excluded if they had any of the following: (1) an unstable medical situation; (2) a prior history of cognitive impairment; (3) communication or speech difficulties; or (4) a history of uncontrolled neurological or neurosurgical diseases other than COVID-19. Cases were also excluded if they had a history of COVID-19 prior to the vaccination or if they had persistent headache following vaccination to prevent COVID-19.

### 2.6. Recruitment

To avoid any selection or detection bias, both cases and controls underwent the same screening.

Both cases and controls were attended to in primary care. Purposive non-probability sampling was conducted so that researchers could screen patients with suspected COVID-19.

Eligibility criteria were confirmed by the research team, both general practitioners and neurologists. All eligible patients were invited to participate.

### 2.7. Intervention

Both cases and controls underwent the same evaluation and follow-up. A standardized hetero-administered questionnaire was dispensed by a trained healthcare worker if they met the eligibility criteria. The questionnaire was adapted from prior studies [1]. Patient evaluation was performed in both groups within 7–14 days from the symptom onset, to minimize recall bias. All patients were prospectively followed up at the headache unit for a minimum of three months by clinical interviews.

### 2.8. Variables

The study questionnaire included parameters related to patients’ demographics, prior medical history, clinical presentation of COVID-19, headache duration and phenotype, and acute treatment of headache. Demographic variables included age, sex, and race. Prior medical history variables were a history of hypertension, diabetes, smoking status, cardiac disorders, respiratory disorders, oncological diseases, or immunosuppression. The full definition of the comorbidities is available in the Appendix A. Prior headache history addressed the prior personal and family history of headache disorders. In relation to headache, the duration (in days), localization, quality of pain, intensity (numeric rating scale, where 0: no pain; 10: worst possible pain), associated symptoms, and worsening by physical activity were investigated. The need for the acute treatment of the headache during the COVID-19 acute phase was assessed, including the specific drugs that were used and the lack of response to acute treatment, as per patients’ subjective opinion.

### 2.9. Statistical Analysis

Qualitative and ordinal variables are presented as frequency and percentage. Quantitative variables are reported as the mean and standard deviation (SD) or median and inter-quartile range (IQR), depending on the distribution. Normality was assessed with the Kolmogorov–Smirnov test. Homoscedasticity or heteroscedasticity was assessed with Levene’s test. In the hypothesis test between qualitative variables, Fisher’s Exact test was used. In the case of quantitative variables, an independent samples Student’s T test or Mann–Whitney U test was used, depending on the distribution. The Log Rank test was used to compare the duration of the headache, presenting data as the hazard ratio (HR) and 95% confidence interval (CI). The statistical significance threshold was set at 0.05. In the event of multiple comparisons, the False Discovery Rate procedure was applied [10]. Missing data were managed with complete case analysis. There was no sample size estimation, and the analyses were performed on the available data. Statistical analysis was performed with the Statistical Package for Social Sciences (SPSS^®^, version 25.0, IBM Corp. Armonk, NY, USA).

## 3. Results

During the study period, 1790 patients were screened, and 453 met the eligibility criteria, including 350 non-vaccinated and 103 vaccinated patients that visited a primary care setting due to suspected COVID-19 disease and were managed in an outpatient setting. A flow diagram is shown in Figure 1.

### 3.1. Demographic Variables and Prior Medical History

Table 1 summarizes the main demographic variables. The median age was 42 years, and 69% were female. There were only statistically significant differences regarding the proportion of patients with a prior family history of headache.

### 3.2. Clinical Presentation of COVID-19

The most frequently reported symptoms during the acute phase of COVID-19 were asthenia (83.2%), weakness (65%), cough (64.7%), and fever (60.8%). Anosmia, ageusia, diarrhea, and cutaneous rash were less frequent in vaccinated patients, while myalgia, arthralgia, expectoration, and rhinorrhea were more frequent in vaccinated patients (Table 2).

### 3.3. Duration of Headache

Headache was the first experienced symptom during the acute phase of COVID-19 in 103 (33.3%) patients, both in vaccinated and non-vaccinated patients (35% vs. 32.5%, *p* = 0.670). Headache had a shorter duration in vaccinated patients (4 (IQR 2–8) vs. 8 (IQR 4–16.5) days, *p* < 0.001). Figure 2 represents the Kaplan–Meier curve of the duration of headache over time. At month 3, 2 (1.9%) vaccinated patients and 14 (6.8%) unvaccinated patients still experienced headache. A prior history of headache, migraine, or tension-type headache was not associated with a different duration of headache (*p* > 0.05, Appendix A).

### 3.4. Headache Phenotype

The most frequent headache phenotype in the acute phase of COVID-19 was holocranial, with a predominantly frontal location, pressing in quality, and of severe intensity. In vaccinated patients, rhinorrhea, nasal congestion, and sweating were more frequent (Table 3).

### 3.5. Acute Treatment of Headache

Ninety-five percent of patients needed the acute treatment of headache. Table 4 shows the frequency of use of the main acute medications. The median number of employed treatments was one (IQR 1–2). The number of acute medications employed was higher in vaccinated patients (Figure 3). Paracetamol was the most frequently used drug. Non-steroidal anti-inflammatory drugs (NSAIDs) were more frequently used in vaccinated patients than in non-vaccinated patients (*p* < 0.001). Patients who did not use acute medication had a shorter duration of the headache (HR: 0.327; 95%CI: 0.186–0.576; *p* < 0.001). The use of NSAIDs was not associated with a shorter/more prolonged duration of the headache (HR: 0.953; 95%CI: 0.751–1.211; *p* = 0.695) (Appendix A). Treatment resistance was reported by 81 (26.2%) patients, including 37 (35.9%) vaccinated patients and 44 (21.4%) non-vaccinated patients (*p* = 0.006–0.009).

## 4. Discussion

In the present study, the phenotype, including the duration of acute headache attributed to COVID-19, was compared between vaccinated and non-vaccinated patients. Regarding headache duration, the distribution of headache days in both populations differs according to vaccination status.

Headache in the acute COVID-19 phase in vaccinated patients resulted in a more frequent presentation of certain cranial autonomic symptoms. In addition, multiple symptoms were more frequent in vaccinated patients, including myalgia, arthralgia, expectoration, and rhinorrhea, while anosmia, ageusia, and diarrhea were less frequent in vaccinated individuals.

Since in previous studies, non-vaccinated patients were more frequently female, were older, and had a higher frequency of prior headache history than our vaccinated group [1], vaccinated patient cases were paired with controls that were matched for the three parameters. This is particularly relevant since they were associated with a more prolonged duration of headache in previous studies [3]. The only difference between both groups was the frequency of a family history of headache, which was two-fold higher in vaccinated patients. Patients with a prior history of headache disorders, mainly migraine, have an increased risk of developing primary headache disorders. Nevertheless, in our study, the duration of acute headache related to COVID-19 infection lasted for a median of four days in vaccinated patients, compared to eight days in non-vaccinated patients.

Our findings present an additional reason for recommending vaccination to prevent COVID-19 while also considering the reported decrease in the mortality and severity of COVID-19 [11,12,13]. Some studies had previously suggested that vaccinated patients had 40% less probability of having headache during COVID-19, compared to non-vaccinated patients [7]. The added value of our study shows that headache is not only less frequent but also shorter in duration in this case. This is particularly relevant, since patients who present headache describe it as the most bothersome symptom [1], and despite being brief, headache was treatment-resistant in a third of vaccinated individuals and a fifth of non-vaccinated individuals. To date, the acute and preventive treatment of COVID-19-related headache is based on the clinical phenotype, with no specific therapies so far [6]. Though we cannot undoubtedly probe the way that symptomatic treatment affected the headache timeline, there are data supporting the influence of vaccination on headache duration. So, the need for acute medication was comparable between vaccinated and non-vaccinated patients, and NSAID use was higher among non-vaccinated patients. The use of NSAIDs, however, was not associated with a different headache duration, and treatment resistance was higher among vaccinated patients, but despite this, headache duration was shorter.

The frequency of acute medication use differed between vaccinated and non-vaccinated patients; however, 96% and 94% of patients used at least one acute treatment. This could be related to the safety concerns that arose during the first wave of the pandemic, when the safety of NSAIDs was not sufficiently established. This is supported by the finding of these drugs being less used by non-vaccinated individuals, which were recruited at that time.

On the one hand, headache is the most common symptom following vaccination [11,12,13]. It typically occurs within the first 72 h, and it lasts for 24–96 h [14,15]. Its pathophysiology has been linked to the immune response triggered by immunization [16]. On the other hand, COVID-19-related headache has also been associated with the immune response and cytokine and interleukin release [17]. This partially explains the better prognosis of COVID-19 patients who had headache as an onset symptom, compared to patients without headache [4,5,16,17,18]. In vaccinated patients, the immune response to infection should occur earlier, and it would include both innate and cellular immunity [19].

Another finding that would support a different immune response is the clinical presentation of COVID-19. Vaccinated patients had a higher frequency of myalgia and arthralgia, which have been associated with a boosted immune response against the virus [20]. This could explain the more frequent treatment resistance to acute medications for those showing a briefer but more difficult to treat headache. There were also differences regarding anosmia, ageusia, and upper respiratory tract symptoms, such as rhinorrhea or expectoration. This could be related to the specific variants, as well.

Some of the main limitations of the current study were the absence of the microbiological determination of the specific variant and that cases and controls were recruited at different times. Non-vaccinated patients were recruited during the first wave of the pandemic, where only the historical/Wuhan variant circulated in Spain [21]. In the case of vaccinated patients, Omicron was isolated in more than 98% of infections according to country-level estimates [22]. Therefore, the prevalence and duration of headache in vaccinated individuals should be within the range of those of patients infected by the Omicron variant. Existing evidence regarding the impact of different variants on COVID-19-related headache is more related to the prevalence of headache during the acute phase than to the duration of the headache [23]. A national-level study performed in the United Kingdom reported that headache was less frequently reported by patients infected with the Omicron variant [24]. Another Turkish study that compared the historical/Wuhan variant (*n* = 960) with the Omicron variant (*n* = 411) found that the prevalence of headache was higher in patients infected by Omicron (7.6% vs. 12.7%) [25]. The only headache-specific study compared the prevalence of headache both during the acute phase and six months after the acute phase in patients infected by the historical/Wuhan variant (*n* = 201), alpha variant (*n* = 211), or delta (*n* = 202) variant. Headache prevalence as an onset COVID-19 symptom was 20.9%, 11.8%, and 32.7%, respectively, and the prevalence of headache six months after the acute phase was 5.5%, 3.8%, and 12.9%, respectively [1]. Hence, it seems that patients affected by the Omicron variant present the highest risk of headache during the acute phase. In our study, the presence of a new-onset headache experienced during COVID-19 was an inclusion criterion, and therefore, this study was not impacted by the risk of developing headache, depending on the headache variant, but by the duration of the headache. In this regard, a recent meta-analysis that included 51 studies and 33,573 patients showed that the prevalence of headache, three months after the acute phase, was the highest in patients infected by the alpha variant (65.8%; 95% CI: 47.7–83.9%), followed by the Omicron variant (52.1%; 95% CI: 44.0–60.1%), Beta variant (34.6%; 95% CI: 27.2–41.9%), delta variant (28.4%; 95% CI: 7.9–49.0%), and historical/Wuhan variant (10.0%; 95%: 7.6–12.4%) [26]. So, in our study, the observed three-month prevalence of headache in unvaccinated patients was in line with that observed in the literature, but there was a much lower prevalence of headache in vaccinated individuals, suggesting that immunization may have a very relevant role in the risk of developing persistent headache after COVID-19 resolution. This may pose a relevant limitation, but it seems difficult to study it otherwise in the present conditions, due to the following reasons: (a) the proportion of patients affected by the Wuhan variant seems very low currently; (b) when the Wuhan variant was circulating, vaccines were not available yet; (c) when the Omicron variant emerged, vaccination campaigns had begun, and in our setting, the vaccine coverage reached three quarters of the population [22]; and (d) conducting this study afterwards could imply that patients had previously been in contact with the virus, which could imply some degree of immune system boosting too.

Other limitations included the absence of hospitalized patients or cases managed in an inpatient setting, where our findings could not necessarily be generalized. Concerning external validity, our sample was composed mainly of Caucasian patients, while other populations were underrepresented. Some strengths of this study are the confirmed diagnosis of COVID-19, the presence of prospective follow-up, and the specific evaluation of patients, using a standardized questionnaire. In addition, we lack a negative control group of noninfected patients and those with background headaches.

And a final limitation to point out is that we did not control all variables related to symptomatic treatment and their influence on headache resolution. Neither during the first phase (unvaccinated patients) nor in the second one (vaccinated) was there enough evidence regarding recommended symptomatic therapy for COVID-19-related headache, so we controlled the need for such treatments and the number of drugs used.

Finally, we understand that the process of generalizing these results, especially regarding the significance of the relationship between vaccination and the duration of headache, is somewhat speculative. The fact that many of these patients suffer from primary headache means that the presence of an infection, which may worsen this condition or cause a new headache to appear, introduces a bias that we were not able to resolve with the design of our study.

## 5. Conclusions

Patients who suffered from COVID-19 after an adequate vaccination presented a different clinical phenotype with a higher frequency of cranial autonomic symptoms than non-vaccinated patients. These also presented a lower frequency of anosmia, ageusia, diarrhea, and rash and a higher frequency of myalgia, arthralgia, expectoration, and rhinorrhea. The duration of headache among vaccinated patients is shorter, although the design of this study does not allow us to determine the different influences of the vaccine, symptomatic treatment, or the variant of COVID-19 on this finding.

## Figures and Tables

**Figure 1 biomedicines-13-02900-f001:**
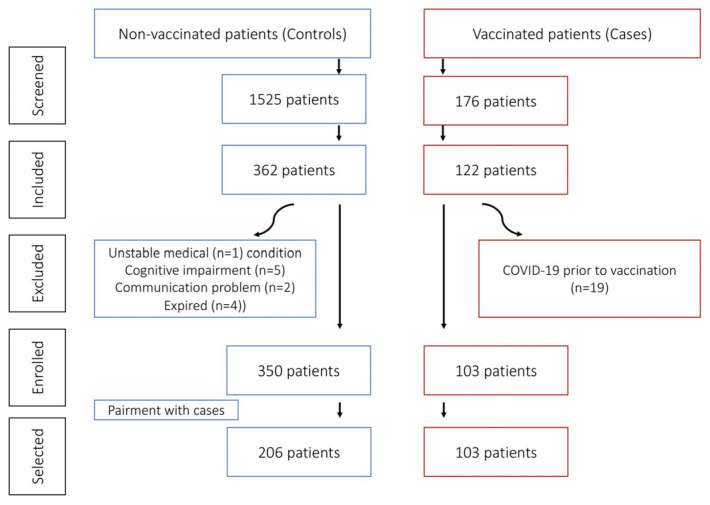
Flow diagram of screened, included, and excluded patients.

**Figure 2 biomedicines-13-02900-f002:**
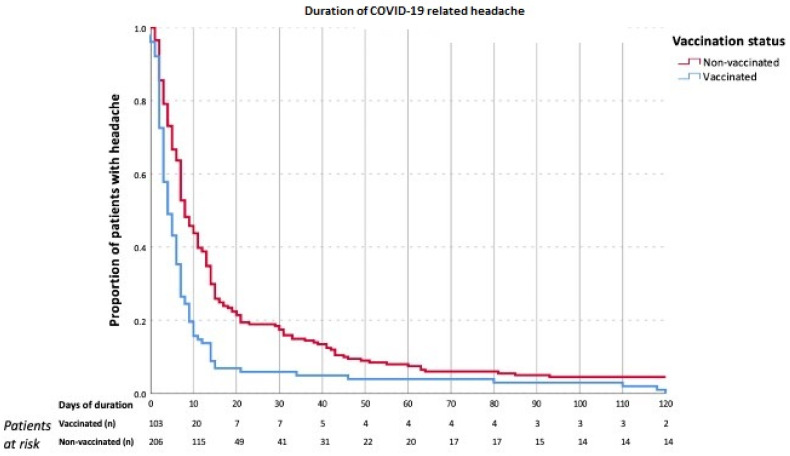
Duration of headache (days), comparing vaccinated (light blue) and non-vaccinated (red) patients (*p* < 0.001, Log Rank test).

**Figure 3 biomedicines-13-02900-f003:**
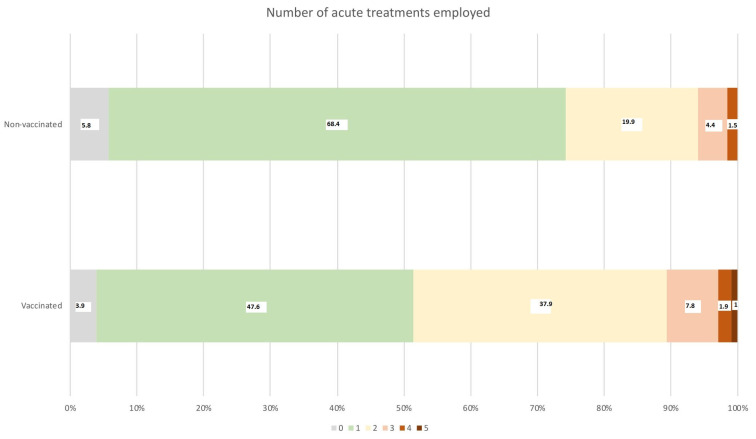
The proportion of patients that used 0-1-2-3-4 or 5+ acute treatments in the management of COVID-19-related headache.

**Table 1 biomedicines-13-02900-t001:** Demographic variables and prior medical history of patients.

Variable	Entire Study Sample (*n* = 309)	Vaccinated Patients (*n* = 103)	Non-Vaccinated Patients (*n* = 206)	FDR-Adjusted *p*-Value
Median age	42 (IQR 33–51)	42 (IQR 33–51)	42 (IQR 34–52)	0.999
Female sex	213 (68.9%)	73 (70.9%)	140 (68.0%)	0.999
Caucasian	280 (90.6%)	97 (94.2%)	183 (88.8%)	0.981
Hypertension	27 (8.7%)	8 (7.8%)	19 (9.2%)	0.900
Diabetes	17 (5.5%)	5 (4.9%)	12 (5.8%)	0.943
Active smokers	35 (11.3%)	15 (14.6%)	20 (9.7%)	0.822
Prior history of cardiac disorders	11 (3.6%)	3 (2.9%)	8 (3.9%)	0.984
Prior history of pulmonary disorders	25 (8.1%)	10 (9.7%)	15 (7.3%)	0.999
Oncologic disorders	5 (1.6%)	0 (0%)	5 (2.4%)	0.749
Immunosuppression	1 (0.3%)	1 (1.0%)	0 (0%)	0.865
Prior history of headache	81 (26.2%)	26 (25.2%)	55 (26.7%)	0.891
Prior history of migraine	40 (12.9%)	17 (16.5%)	23 (11.2%)	0.210
Prior history of tension-type headache	31 (10.0%)	9 (8.7%)	22 (10.7%)	0.690
Prior history of acute headache with other prior viral infection	101 (31.7%)	36 (35%)	65 (31.6%)	0.984
Family history of headache	83 (26.9%)	46 (44.7%)	37 (18.0%)	<0.001

**Table 2 biomedicines-13-02900-t002:** The frequency of reported symptoms during the acute phase of COVID-19.

Variable	Entire Study Sample (*n* = 309)	Vaccinated Patients (*n* = 103)	Non-Vaccinated Patients (*n* = 206)	FDR-Adjusted *p*-Value
Asthenia	257 (83.2%)	91 (88.3%)	166 (80.6%)	0.180
Weakness	201 (65.0%)	74 (71.8%)	127 (61.7%)	0.147
Cough	200 (64.7%)	73 (70.9%)	127 (61.7%)	0.200
Fever	188 (60.8%)	59 (57.3%)	129 (62.6%)	0.412
Myalgias	183 (59.2%)	75 (72.8%)	108 (52.4%)	0.004
Arthralgias	147 (47.6%)	60 (58.3%)	87 (42.2%)	0.03
Anosmia	135 (43.7%)	19 (18.4%)	116 (56.3%)	<0.001
Ageusia	116 (37.5%)	14 (13.6%)	102 (49.5%)	<0.001
Dyspnea	109 (35.3%)	30 (29.1%)	79 (38.3%)	0.184
Dizziness	114 (25.2%)	31 (30.1%)	83 (23.7%)	0.229
Chest pain	73 (23.6%)	24 (23.3%)	49 (23.8%)	0.999
Rhinorrhea	97 (31.4%)	68 (66%)	29 (14.1%)	<0.001
Diarrhea	96 (31.1%)	19 (18.4%)	77 (37.4%)	0.0031
Odynophagia	94 (30.4%)	27 (26.2%)	67 (32.5%)	0.334
Lightheadedness	77 (24.9%)	31 (30.1%)	46 (22.3%)	0.213
Expectoration	65 (21.0%)	30 (29.1%)	35 (17.0%)	0.038
Cutaneous rash	30 (9.7%)	4 (3.9%)	26 (12.6%)	0.034

**Table 3 biomedicines-13-02900-t003:** Variables related to the headache phenotype.

Variable	Entire Study Sample (*n* = 309)	Vaccinated Patients (*n* = 103)	Non-Vaccinated Patients (*n* = 206)	FDR-Adjusted *p*-Value
Holocranial location *	256 (82.6%)	92 (89.3%)	164 (79.6%)	0.134
Hemicranial location *	51 (16.5%)	11 (10.7%)	40 (19.4%)	0.139
Frontal topography	149 (48.2%)	54 (52.4%)	95 (46.1%)	0.569
Temporal topography	88 (28.5%)	33 (32%)	55 (26.7%)	0.566
Parietal topography	40 (12.9%)	16 (15.5%)	24 (11.7%)	0.565
Occipital topography	46 (14.9%)	19 (18.4%)	27 (13.1%)	0.429
Periocular topography	48 (15.5%)	17 (16.5%)	31 (15.0%)	0.826
Vertex topography	17 (5.5%)	5 (4.9%)	12 (5.8%)	0.857
Cervical topography	15 (4.9%)	2 (1.9%)	13 (6.3%)	0.325
Pressing quality	238 (77.0%)	85 (82.5%)	153 (74.3%)	0.280
Throbbing quality	40 (12.9%)	14 (13.6%)	26 (12.6%)	0.858
Stabbing quality	51 (16.5%)	21 (20.4%)	30 (14.6%)	0.382
Electric quality	4 (1.3%)	2 (1.9%)	2 (1.0%)	0.794
Burning quality	5 (1.6%)	1 (1%)	4 (1.9%)	0.807
Intensity of headache in NPRS	7 (IQR 6–8)	7 (IQR 6–8)	7 (IQR 6–8)	0.810
Photophobia	88 (28.5%)	31 (30.1%)	57 (27.7%)	0.799
Phonophobia	109 (35.3%)	46 (44.7%)	63 (30.6%)	0.098
Osmophobia	15 (4.9%)	6 (5.8%)	9 (4.4%)	0.845
Clinophilia	201 (65.0%)	57 (55.3%)	144 (69.9%)	0.116
Worsening by physical activity	98 (31.7%)	35 (34.0%)	63 (30.6%)	0.761
Nausea	45 (14.6%)	10 (9.7%)	35 (17.0%)	0.274
Vomit	19 (6.1%)	7 (6.8%)	12 (5.8%)	0.831
Sweating	46 (14.9%)	29 (28.2%)	17 (8.3%)	<0.001
Red eye	27 (8.7%)	14 (13.6%)	13 (6.3%)	0.150
Tearing	31 (10.0%)	16 (15.5%)	15 (7.3%)	0.116
Rhinorrhea	39 (12.6%)	30 (29.1%)	9 (4.4%)	<0.001
Nasal congestion	64 (20.7%)	45 (43.7%)	19 (9.2%)	<0.001
Ptosis	5 (1.6%)	4 (3.9%)	1 (0.5%)	0.141
Otic plenitude	31 (10.0%)	16 (15.5%)	15 (7.3%)	0.135

NPRS: Numeric pain rating scale. * Two patients reported headache located in the midline with no lateralization.

**Table 4 biomedicines-13-02900-t004:** Frequency of use of each acute medication group.

Variable	Entire Study Sample (*n* = 309)	Vaccinated Patients (*n* = 103)	Non-Vaccinated Patients (*n* = 206)	FDR-Adjusted *p*-Value
Need for acute medication	293 (94.8%)	99 (96.1%)	194 (94.2%)	0.999
Paracetamol	262 (84.8%)	84 (81.6%)	178 (84.6%)	0.939
NSAIDs	104 (33.7%)	59 (57.3%)	45 (21.8%)	<0.001
Metamizole	46 (14.9%)	16 (15.5%)	30 (14.6%)	0.999
Triptan	5 (1.6%)	2 (1.9%)	3 (1.5%)	0.999
Tramadol	2 (0.6%)	1 (1.0%)	1 (0.5%)	0.999

NSAIDs: Non-steroidal anti-inflammatory drugs.

## Data Availability

Datasets and additional information are available upon reasonable request to the corresponding author.

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
