# Peer review of "COVID-19 Vaccination Modifies COVID-19-Related Headache Phenotype: Evidence from Case–Control Study on 309 Participants"

_biomedicines, 2025, doi:10.3390/biomedicines13122900_

Round 1

Reviewer 1 Report (Previous Reviewer 1)

Comments and Suggestions for Authors

The revised manuscript was improved according to the  reviewers' recommendations. 

Author Response

Enclose letter

We have thoroughly revised the manuscript following the recommendations made by the Reviewers.
We believe that the changes and additions to the manuscript in this new version have significantly improved the overall quality of the paper.
The changes in the text have been highlighted in yellow.

Author reply to reviewer no.1 comment: We appreciate your comments.

Reviewer 2 Report (Previous Reviewer 3)

Comments and Suggestions for Authors

The manuscript is improved 

Author Response

Enclose letter

We thank again the Reviewers for their feedback on our revised manuscript.
We have thoroughly revised the manuscript following the recommendations made by the Reviewers.
We believe that the changes and additions to the manuscript in this new version have significantly improved the overall quality of the paper.
The changes in the text have been highlighted in yellow.

Author reply to reviewer no.2 comment: Thanks for your recommendations.

Reviewer 3 Report (Previous Reviewer 2)

Comments and Suggestions for Authors

Paper “COVID-19 vaccination decreases COVID-related headache duration: Evidence from a case-control study on 309 participants” by Henar Ruiz-Saez and coauthors covers the results of a clinical trial on COVID-19-related headaches involving 309 participants and was previously considered for publication in Biomedicines with decision to reject publication due to serious issues that must be addressed. The revised manuscript contains changes to address four issues that were pointed out to. Despite authors were suggested to resolve major issues regarding building evidence and communication of results, only minor excuses to study limitation were introduced. Therefore the manuscript still contains not resolved issues that were initially raised. Below are the comments on the author’s response letter.

  1. Table 2. Table now contains data with the results for observed symptoms. Resolved.
  2. Headache treatment monitoring. Authors apparently cannot provide data that prove unbiased nature of conclusions of authors that vaccination improves headache conditions during COVID-19 infection. Therefore, I suggest authors to change the main message of manuscript (COVID-19 vaccination decreases COVID-related headache duration) to not mislead readers as authors cannot prove that headache treatment regimen does not affect headache resolution timeline. This change requires a substantial revision of the entire manuscript, not just the comments about the limitations of the study.
  3. Negative control group. Authors highlighted in response letter that there is no significant difference between vaccinated/non-vaccinated groups in terms of primary headache. However it worth also note that 26.2% of all recruited patients had primary headache. This fraction of population is not negligible. Inclusion of negative control group could help to show background headache conditions, but it is not the only option. Without negative control group authors should provide the statistical analysis to prove that sample size is enough then provide the estimated level of confidence to draw the conclusions that are summarized in title of manuscript.
  4. Generalization of findings on headache duration for vaccinated. It should be noted one more time that authors do not communicate appropriately the implications of study. The discussion of findings in context of population size of vaccinated and then infected people on the scale of overall population of vaccinated should be added.

Author Response

Enclose letter

Round 2

Reviewer 3 Report (Previous Reviewer 2)

Comments and Suggestions for Authors

Paper “COVID-19 Vaccination Modifies COVID-Related Headache Phenotype: Evidence from a Case-Control Study on 309 Participants” by Henar Ruiz-Saez and coauthors covers the results of a clinical trial on COVID-19-related headaches involving 309 participants and was previously considered for publication in Biomedicines with decision to reject publication due to serious issues that must be addressed. The revised manuscript contains changes to address the issues, but a number of improvements are still needed.

  1. Headache treatment monitoring. Acknowledging in the discussion section the inability to prove a reduction in headache duration based on the available data makes the article less misleading. Furthermore, it is suggested to include relevant comments in the conclusions.
  2. Negative control group. The authors decline to provide statistical analysis confirming the adequacy of the sample size and subsequently indicate the assumed confidence level for the conclusions stated in the manuscript title. The reviewer suggests at least adding a discussion of the statistical significance of the shorter headache duration in vaccinated patients (4 (interquartile range 2–8) vs. 8 (interquartile range 4–16.5) days, p < 0.001) to ensure an objective presentation of the results.
  3. Generalization of findings on headache duration for vaccinated. Vaccination-associated headache has nothing to do with the reviewer's comment. In fact, it would be worth adding sentences that explicitly describe the magnitude of the study's implications taking into account the population sizes of unvaccinated and infected, vaccinated and infected, vaccinated and uninfected, and unvaccinated and uninfected people.

Author Response

PDF attached

This manuscript is a resubmission of an earlier submission. The following is a list of the peer review reports and author responses from that submission.

Round 1

Reviewer 1 Report

Comments and Suggestions for Authors

The manuscript addresses an important and clinically relevant question. Nevertheless, the definition of the groups raises concerns about potential analytical bias. The description of the matching process lacks sufficient detail, particularly regarding recruitment procedures and possible pre-selection of participants. Providing a clearer explanation of these aspects would enhance the robustness and transparency of the study.

Reviewer 2 Report

Comments and Suggestions for Authors

Paper “COVID-19 vaccination decreases COVID-related headache duration: Evidence from a case-control study on 309 participants” by Henar Ruiz-Saez and coauthors covers the results of a clinical trial on COVID-19-related headaches involving 309 participants. The paper is well-structured, but the presentation of data was negligent, as important pieces of evidence were not provided. The paper should be revised prior to consideration for publication. Following issues should be addressed.

  1. Table 2, expected to show the results for observed symptoms was replaced with previously presented Table 1 containing demographic data, so the reviewer cannot verify if the described results are supported by the evidence.
  2. One major drawback of the study design is that headache treatment was not monitored closely enough. Only the number of treatments was counted, but temporal parameters and connections with other treatments or symptoms were not collected. These parameters could be crucial for understanding the variation in headache resolution times in vaccinated and non-vaccinated groups, which is the main finding of the paper! However, the authors ignored these parameters. Data proving that treatment type and regimen do not bias the author's conclusions should be included.
  3. Another major gap in the study is the lack of a negative control group consisting of patients who were not infected, but had background headache. Results of the study (e.g., differences in headache symptom resolution) should be statistically verified not only for the groups that were infected and had symptoms associated with it, but should also include data on the background incidence of headaches and variation in the overall population.
  4. Another flaw in the study was the absence of a discussion about the fact that vaccinated and infected individuals had an immune response to the vaccine that was unable to protect them from infection (vaccine failure). Given the high effectiveness of vaccines, this group of vaccinated and infected people is a minority of the vaccinated population. Therefore, the claimed findings of the paper that vaccination reduces COVID-19 headaches cannot be generalized and have limited significance. The implications of the research should be more clearly communicated.

Reviewer 3 Report

Comments and Suggestions for Authors

The manuscript threads on a thin line of the headache related to COVID. The protocol is based on an observation design, with numerous problem sin methodological approach and analysis. Firstly, the sample size is too small for such an effect to be detected decently. I think this is mostly a false positive result, caused by the sample composition, measurements and analytic approach. Table 3 shows no difference in headache frequency, so it remains unclear why the authors progressed beyond this stage. Next, the composition of cases and controls is uneven, and there is no adjusting to that fact by the use of regression methods. Semi-automated matching sounds dangerous. Next, headache is a subjective phenomenon, and this could all simply fit in the placebo effect, where the vaccinated people were more likely to believe that they were covered, vs. mass hysteria od the unvaccinated who got the disease, which was especially prevalent during the sampling periods (mind you, why would you have different recruitment time frames for cases and controls, or even, why mention this)? Therefore, I find the manuscript severely flawed in methodological terms. If the authors would like to resubmit anyhow, then they must treat this in a more serious and more hypothesis generating way, provide claims that this is just a suggestion and not a reliable evidence, and they need to make sure to incorporate a number of methodological limitations – a simple regression model with age, sex and comorbidities (binary) might provide a more reliable answer, and that should not be very hard to produce.